pH-sensing G protein-coupled orphan receptor GPR68 is expressed in human cartilage and correlates with degradation of extracellular matrix during OA progression

Khan Nazir M. mnkhan2@emory.edu
Diaz-Hernandez Martha E.
http://orcid.org/0000-0002-6023-8198 Martin William N.
Patel Bhakti
Chihab Samir
Drissi Hicham
Orthopaedics, Emory University , Atlanta, GA , United States
Pan Feng
Electronic publication date: 2023 Dec 5
Publication date: 2023
Volume: 11
Electronic Location ID: e16553
Received 2023 May 7; Accepted 2023 Nov 9
Copyright: © 2023 Khan et al.
Copyright year: 2023
Copyright holder: Khan et al.
License: This is an open access article distributed under the terms of the Creative Commons Attribution License, which permits unrestricted use, distribution, reproduction and adaptation in any medium and for any purpose provided that it is properly attributed. For attribution, the original author(s), title, publication source (PeerJ) and either DOI or URL of the article must be cited.
License URL: https://creativecommons.org/licenses/by/4.0/

Keywords: Orphan receptor, Chondrocytes, Osteoarthritis, GPR68, Matrix degeneration

Funding: NIH-National Center for Advancing Translational Sciences (NCATS) R03TR003669-01A1 Emory University to Hicham Drissi This work was supported by the NIH-National Center for Advancing Translational Sciences (NCATS) (R03TR003669-01A1) to Nazir M Khan and Funds from Emory University to Hicham Drissi. The funders had no role in study design, data collection and analysis, decision to publish, or preparation of the manuscript.

==============================
Background

Osteoarthritis (OA) is a debilitating joints disease affecting millions of people worldwide. As OA progresses, chondrocytes experience heightened catabolic activity, often accompanied by alterations in the extracellular environment’s osmolarity and acidity. Nevertheless, the precise mechanism by which chondrocytes perceive and respond to acidic stress remains unknown. Recently, there has been growing interest in pH-sensing G protein-coupled receptors (GPCRs), such as GPR68, within musculoskeletal tissues. However, function of GPR68 in cartilage during OA progression remains unknown. This study aims to identify the role of GPR68 in regulation of catabolic gene expression utilizing an in vitro model that simulates catabolic processes in OA.

Methods

We examined the expression of GPCR by analyzing high throughput RNA-Seq data in human cartilage isolated from healthy donors and OA patients. De-identified and discarded OA cartilage was obtained from joint arthroplasty and chondrocytes were prepared by enzymatic digestion. Chondrocytes were treated with GPR68 agonist, Ogerin and then stimulated IL1β and RNA isolation was performed using Trizol method. Reverse transcription was done using the cDNA synthesis kit and the expression of GPR68 and OA related catabolic genes was quantified using SYBR® green assays.

Results

The transcriptome analysis revealed that pH sensing GPCR were expressed in human cartilage with a notable increase in the expression of GPR68 in OA cartilage which suggest a potential role for GPR68 in the pathogenesis of OA. Immunohistochemical (IHC) and qPCR analyses in human cartilage representing various stages of OA indicated a progressive increase in GPR68 expression in cartilage associated with higher OA grades, underscoring a correlation between GPR68 expression and the severity of OA. Furthermore, IHC analysis of Gpr68 in murine cartilage subjected to surgically induced OA demonstrated elevated levels of GPR68 in knee cartilage and meniscus. Using IL1β stimulated in vitro model of OA catabolism, our qPCR analysis unveiled a time-dependent increase in GPR68 expression in response to IL1β stimulation, which correlates with the expression of matrix degrading proteases suggesting the role of GPR68 in chondrocytes catabolism and matrix degeneration. Using pharmacological activator of GPR68, our results further showed that GPR68 activation repressed the expression of MMPs in human chondrocytes.

Conclusions

Our results demonstrated that GPR68 was robustly expressed in human cartilage and mice and its expression correlates with matrix degeneration and severity of OA progression in human and surgical model. GPR68 activation in human chondrocytes further repressed the expression of MMPs under OA pathological condition. These results identify GPR68 as a possible therapeutic target in the regulation of matrix degradation during OA.

Introduction

Osteoarthritis (OA) is a debilitating degenerative joint disease and causes a severe economic and social burden (Goldring, 1999). OA is characterized by focal articular cartilage degeneration, osteophyte formation, degradation of extracellular matrix, and changes in the subchondral bone (Goldring & Berenbaum, 2004; Loeser, Collins & Diekman, 2016). Matrix degradation and cartilage catabolism are important component of OA pathogenesis, which is mediated by induced expression of matrix degrading proteases such as MMPs and ADAMTSs (Goldring, 1999; Goldring & Berenbaum, 2004; Goldring & Marcu, 2009). Several in vitro and in vivo studies have implicated the role of pro-inflammatory cytokine interleukin (IL)1β in degeneration of extracellular matrix in OA (Goldring et al., 2008; Sokolove & Lepus, 2013). The target cell of IL1β is the chondrocyte, which in OA cartilage shows a dysregulated expression of catabolic and anabolic genes, resulting in imbalanced homeostasis (Goldring & Marcu, 2009; Goldring et al., 2008). IL1β stimulates the expression of matrix degrading proteases (MMPs and ADAMTS) in chondrocytes and cartilage explants in vitro and is believed to be a critical mediator of OA pathogenesis (Goldring, 1999; Haseeb & Haqqi, 2013; Malemud, Islam & Haqqi, 2003). To date, no disease-modifying drug has been identified to treat or reverse OA progression. Thus, efforts to identify new molecular targets that can modulate the matrix degeneration and cartilage catabolism is of utmost importance.

During OA progression, chondrocytes undergo higher catabolism which often accompanied by fluctuations in extracellular osmolarity and acidity (Christensen et al., 2005; Collins et al., 2013; Hall, Horwitz & Wilkins, 1996; Roman et al., 2017). These changes in OA joints are manifested due to increased proton leakage and decreased extracellular pH which results in imbalance in pH homeostasis leading to acidosis of synovial fluid and the extracellular environment of articular chondrocytes (Christensen et al., 2005; Collins et al., 2013; Hall, Horwitz & Wilkins, 1996; Roman et al., 2017). However, the mechanism by which chondrocytes sense and respond to acidotic stress and defend against the detrimental effects of acidification and the regulatory proteins involved in these defenses is not known. Recently, roles of a family of pH-sensing G protein-coupled receptors (GPCRs), including GPR4, GPR65, G2A, and GPR68 have recently become of interest in musculoskeletal tissues (Yang et al., 2006; Pereverzev et al., 2008; Tomura et al., 2008; Yuan et al., 2014a, 2014b). Our initial transcriptome analysis unveiled the presence of all four members of pH-sensing receptors in both human and murine cartilage. However, when we conducted a comparative expression analysis, it became evident that among these receptor members, GPR68 exhibited notably higher expression levels. Moreover, GPR68 displayed a significant increase in expression within OA cartilage compared to its healthy counterpart, implying a potential role for GPR68 in the development of OA. Consequently, the current investigation was formulated to explore the involvement of the pH-sensing receptor GPR68 in the process of cartilage matrix degeneration during the progression of OA, as well as to assess whether it plays a role in the regulation of catabolic gene expression, employing an in vitro model that simulates catabolic processes characteristic of OA.

GPR68 is a proton-sensing orphan receptor involved in pH homeostasis (Ludwig et al., 2003). GPR68 mediates its action by association with G proteins that stimulates inositol phosphate production or Ca2+ mobilization (Ludwig et al., 2003; Yuan et al., 2014a). The receptor is almost silent at pH 7.8 but fully activated at pH 6.8 (Damaghi, Wojtkowiak & Gillies, 2013). A recent study showed that GPR68 is expressed in rat endplate and its expression induced apoptosis in response to acidosis, which further reduced the production of collagen, finally leading to degeneration of the intervertebral disc (Yuan et al., 2014a). However, its function in articular cartilage and OA progression remains unknown. The present study was designed to study the role of GPR68 in articular cartilage during disease progression using experimental model of OA in human and mice. We next determined the function of GPR68 in the regulation of cartilage matrix gene expression during OA progression. Our study reveals the role of GPR68 in the regulation of matrix degradation during OA thus identifies GPR68 as a possible therapeutic target in management of OA.

Materials and Methods

RNA-sequencing data analysis

To define the role of proton sensing orphan GPCRs in the pathogenesis of OA, we analyzed multiple high throughput RNA-seq data available in public domain (NCBI-GEO) in the cartilage tissue during OA progression. The raw data were downloaded from healthy and OA cartilage tissues in human and mice (GSE106292, GSE110051, GSE114007) available from the NCBI-GEO database (Ferguson et al., 2018; Fisch et al., 2018). The human OA cartilage samples were resected from the weightbearing regions on medial and lateral femoral condyles, and scored as OA grade 4, whereas healthy cartilage samples were of grade 1 and had no history of joint disease or trauma (Fisch et al., 2018). BioJupies (http://biojupies.cloud), a web-based freely available application was used to analyze the differentially expressed genes (DEGs) to determine the expression of proton sensing receptors including GPR68 in healthy vs OA cartilage (Torre, Lachmann & Ma’ayan, 2018). Raw data were normalized to log10 counts per million (log CPM) and differentially expressed genes between the healthy and OA groups were determined using the R package limma (Ritchie et al., 2015). Multiple correction testing was performed using Benjamini-Hochberg method. The Jupyter Notebooks created for each RNA-seq raw data analysis provide differentially expressed gene lists in an interactive data visualizations format (Torre, Lachmann & Ma’ayan, 2018). Mapped read counts of selected genes from Jupyter Notebooks were used for the generation of heatmap using the ClustVis web tool as described previously (Metsalu & Vilo, 2015). The mapped read count was normalized using Z-score for generation of heatmap.

Microarray data analysis

To determine the level of GPR68 in experimental OA, we analyzed the microarray dataset of C57Bl/6 mouse cartilage with DMM and sham operated knee joints. Transcriptome profiling was performed in cartilage samples from mice at 1-, 2- and 6-weeks post-surgery using Agilent-014868 Whole Mouse Genome 4 × 44K G4122F Microarray available from the GEO database (GSE45793) (Bateman et al., 2013). The GEO datasets included RefSeq Gene ID, gene name, Gene symbol, microarray ID, adjusted P-value, and experimental group logFC relative to corresponding control groups. GEO2R was used to analyze the differential expression profiling between sham and DMM operated knee joints at 6-weeks. The FDR method was used to correct for multiple testing in GEO2R. The intensity value was used to represent the fold change of gene expression between sham and DMM operated knee joint cartilage.

450K DNA methylation analysis

To determine the relationship between the correlation of promoter DNA methylation level with gene expression of GPR68, we analyzed global DNA methylation analysis in human OA cartilage samples. To this end, we used publicly available Infinium 450K data sets from 12 knee OA patients (GSE63695) (Steinberg et al., 2017). Genome wide DNA methylation analysis was performed in chondrocytes isolated from two different region of each cartilage- high-grade degeneration (degraded/OA sample) vs low-grade degeneration (intact/healthy sample). All analyses were performed directly on the preprocessed data available online. We used the probe annotations from ChAMP (Morris et al., 2014) to assign promoter probes to gene. After computing the mean beta value in the promoter of GPR68 in each sample, we used a paired t-test to identify differential promoter-region methylation between degraded and intact samples. Significance was set at 5% FDR (false discovery rate).

Preparation of primary human OA chondrocytes

Prior to the initiation of the studies the study protocol was reviewed and approved by the Institutional Review Board (IRB) of Emory University, Atlanta, GA as a “not research involving human subjects” as defined by DHHS and FDA regulations and that no informed consent was needed. All the methods used in this study were carried out in accordance with the approved guidelines and all experimental protocols were approved by the IRB of Emory University, Atlanta, GA (#IRB ID: STUDY00003990). Discarded and de-identified knee joint cartilage samples were collected from the patients undergoing joint arthroplasty at the Emory University Orthopaedics and Spine Hospital (EUOSH), Atlanta, GA.

Macroscopic cartilage degeneration was determined by India ink staining and cartilage degeneration were graded according to Mankin scoring system (Mankin et al., 1971). In this system the scores are defined as follows: 0 = Normal, 1 = Superficial fibrillation, 2 = Pannus and superficial fibrillation, 3 = Fissures to the middle zone, 4 = Fissures to the deep zone and 5 = Fissures to the calcified zone. Cartilage area or sample with high-grade degeneration (Mankin score ≥ 3) signifying severe OA condition were classified as “High-grade OA cartilage”, whereas cartilage area with low-grade degeneration (Mankin score < 3) signifying healthy tissue were classified as “low-grade OA cartilage”. The cartilage surface from knee joint was resected using scalpel blade (Blade #10) and minced into small pieces and chondrocytes were prepared by enzymatic digestion as described previously (Khan et al., 2017a; Khan, Ansari & Haqqi, 2017; Khan, Ahmad & Haqqi, 2018; Khan et al., 2017b, 2017c). Chondrocytes phenotype during in vitro culture was maintained by plating the chondrocytes at high cell density (1 × 106/well of 6-well plate) and phenotype was verified by analyzing the mRNA expression of chondrocytes phenotype markers such as COL2A1, ACAN, and COL10A1 using real time PCR analysis.

In vitro model of OA by stimulation of primary human OA chondrocytes with IL1β

Primary human OA chondrocytes were seeded at high confluency in six well plate (1 × 106/ well) in Dulbecco’s modified Eagle’s medium/Nutrient Mixture F-12 (DMEM/F-12) supplemented with 10% fetal calf serum (FCS), penicillin (100 units/ml), and streptomycin (100 mg/ml) for 2–3 days. When cells reached 80% confluency, OA chondrocytes were serum starved overnight and then stimulated with IL1β (1 ng/ml) for 1, 3, 6, 12 and 24 h at 37 °C, in CO2 incubator under 5% CO2 and 95% air. Unstimulated chondrocytes were used as control.

Ogerin treatment of human chondrocytes and stimulation with IL1β

Human OA chondrocytes cultured in 6-well plate were treated with different concentration of Ogerin (1–10 µM) for 2 h followed by stimulation with IL1β (1 ng/ml) in incomplete DMEM/F-12 medium without presence of serum. Ogerin (OGR) is a positive allosteric modulator (PAM) of GPR68 and known to activate GPR68 at physiological pH. Chondrocytes which were treated with 0.1% DMSO served as control for this experiment.

Total RNA isolation and gene expression analysis using real time qPCR

Chondrocytes were harvested at end of incubation period and total RNA was isolated using TRIzol® method as previously described (Khan, Ahmad & Haqqi, 2018; Khan et al., 2017a; Khan, Ansari & Haqqi, 2017; Khan et al., 2017b, 2020a; Khan et al. 2023a). For gene expression analysis, cDNA was synthesized from 1 µg of total RNA using high-capacity cDNA reverse transcription kit (Life Technologies) and gene expression of GPR68, MMP13, and MMP3 was quantified using SYBR Green expression assay as previously described (Khan, Ahmad & Haqqi, 2018; Khan et al., 2017a; Khan, Ansari & Haqqi, 2017; Khan et al., 2017b; Khan et al. 2023a; Khan et al., 2020a, 2020b, 2023b). β-actin was used as housekeeping gene and relative expression levels were calculated using the 2−ΔΔCT method (Livak & Schmittgen, 2001). Primer sequence for the genes used here were provided in Table 1.

Table 1 Primer sequence for genes used in qPCR analysis.

Gene	Forward primer	Reverse primer	
GPR68	5′-GTCAGTAAATCAACTTCATAGCTCA-3′	5′-AAGGTGGTTCAAGCTCTACC-3′	
MMP3	5′-TGAACAATGGACAAAGGATACAAC-3′	5′-TGAGTGAGTGATAGAGTGGGT-3′	
MMP13	5′-AGCCACTTTATGCTTCCTGA-3′	5-TGGCATCAAGGGATAAGGAAG-3′	

Histological analysis of human OA cartilage

Cartilage matrix depletion due to IL1β-induced activation of matrix degrading proteases was determined by Safranin-O staining of cartilage explants. The cartilage explants stimulated with IL1β (25 ng/ml) for 72 h were fixed in 10% neutral buffer formalin for 24 h, decalcified in 10% EDTA (pH 7.4) for 72 h, dehydrated with series of alcohols and then embedded in paraffin wax, sectioned at 5 μm thickness were stained with Safranin-O dye as described previously (Khan et al., 2017b; Khan et al. 2023b). The cartilage sections were imaged using BioTek Lionheart LX Automated Microscope as described previously (Khan et al., 2020a, 2023a).

Animals and induction of experimental OA using surgical destabilization of the medial meniscus (DMM) in mice

Male C57BL/6 mice were obtained from Jackson Laboratory (ME, USA). Twenty mice were randomly divided into two groups (Sham and DMM) of ten animals per experimental group as described previously (Ansari et al., 2019) and each five mice were kept in one cage. All mice were housed under 12-h light/dark cycle at ambient temperature of 20–25 °C and humidity of 50 ± 10%, and they were given free access to food and water. The animal studies were approved by the Institutional Animal Use and Care Committee of the Atlanta VA medical center (#V002-22). Husbandry and environmental conditions in this facility comply with all governmental regulations. No excess experimental animals were used in this experiment. All mice were euthanized at end of experiment by CO2 asphyxiation, followed by cervical dislocation as secondary physical method of euthanasia.

Experimental OA was induced by performing surgical destabilization of the medial meniscus (DMM) in the left knee of 12 weeks old male mice as described previously (Ansari et al., 2019). Briefly, mice were anesthetized using isoflurane inhalational anesthesia (1.5–2%). The hind limb prepped under sterile conditions and a longitudinal skin incision were made anteriorly on the right knee. The knee joint exposed using a medial parapatellar capsulotomy. The patellar ligament (PL) was laterally displaced, and the knee brought into flexion and medial meniscotibial ligament were dissected with Jewelers forceps to induce destabilization of medial meniscus (DMM). Sham surgery was ed as control. The mice were euthanized at 12-week post-surgery and intact joints were used for the histological assessment of the severity of the induced OA. Knee joints were fixed in a 10% neutral buffered formalin, decalcified in 10% EDTA, embedded in paraffin, and sectioned at 5 µM thickness. Sections were deparaffinized and rehydrated with a series of graded ethanol and stained with Safranin-O/Fast Green to determine loss of proteoglycan content to analyze the loss of cartilage ECM during OA progression.

Immunohistochemical (IHC) analysis of GPR68 in human and mice cartilage

Full thickness human cartilage pieces resected from the human cartilage or intact knee joints harvested 12-week post-surgery from sham and DMM operated mice were fixed in 4% paraformaldehyde for 48 h. Samples were decalcified using 10% EDTA for 2 weeks and dehydrated cartilage pieces were embedded in paraffin and 5 μM thick sections were prepared. Deparaffinization of cartilage sections was performed by incubating histological section with xylene three time (5 min each) and then sections were rehydrated using various grade of alcohol in a decreasing concentration (100%, 90%, 70% and 50%). The sections were washed with 1X TBS for 10 min and antigen retrieval was performed by incubating histological sections in 10 mM citrate buffer (pH 6.0) at 65 °C for 1 h using water bath. To analyze the immunoreactivity of GPR68, the endogenous peroxidase activity was neutralized by 3% hydrogen peroxide in 1X TBS, sections were blocked in 10% normal goat serum for 30 min at room temperature, and incubated in rabbit polyclonal anti-GPR68 antibody (MyBiosource; MBS7044035) overnight at 4 °C. We also used recombinant rabbit IgG monoclonal antibody (Abcam; ab172730) as negative isotype control. Sections were washed with 1X TBS and incubated with HRP-conjugated secondary anti-rabbit antibody for 2 h followed by washing with 1X TBS, and developed using a SignalStain® DAB substrate kit (#8059S; Cell Signaling Technology) and images were captured BioTek Lionheart LX Automated Microscope as described previously (Diaz-Hernandez et al., 2020; Díaz-Hernández et al., 2022; Khan et al., 2023a, 2020a).

Statistical analyses

The data presented here represent the value of Mean ± SD. The statistical difference between control and experimental groups was performed by GraphPad PRISM (GraphPad, San Diego, CA, USA, V 9.0.) using one-way ANOVA with Tukey’s post hoc test. The significance was set at P < 0.05.

Results

Proton-sensing GPCRs are expressed in human and mice cartilage

To examine the role of proton sensing orphan GPCRs in the pathogenesis of OA, we first examined the expression levels of all family members in human and mice cartilage. We analyzed high throughput RNA-seq dataset originally generated for molecular mapping of human skeletal ontogeny and pluripotent stem cell-derived articular chondrocyte (GSE106292) (Ferguson et al., 2018). After stringent quality control, reads per kilobase pair per million mapped reads (RPKM) value (expression) for all four gene members (GRP68, GPR88, GPR132 and GPR4) were determined in human cartilage across four different stages of human ontogeny including (1) embryonic cartilage isolated from limb bud condensations at embryonic 5-6 Week Post Conception (WPC), (2) presumptive cartilage derived from 17 WPC, (3) healthy juvenile cartilage, and (4) adult articular chondrocytes. In addition to human in vivo chondrogenesis, we also analyzed two stages of in vitro pluripotent stem cell (PSC) differentiation into chondrocytes at 14 and 60 days. Our analysis demonstrates that all there four members of proton-sensing orphan GPCR genes expressed in human cartilage, and GPR68 expression was highest in all the human cartilages analyzed (Fig. 1A). We next determined the expression levels of these orphan GPCRs in murine cartilage by further analyzing a different RNA-Seq dataset (GSE110051) generated in immature and mature chondrocytes isolated from the mouse embryo and adult histological section using laser capture microdissection. Our analysis demonstrates variable degree of expression levels of all four GPCRs, with Gpr68 showing highest expression level in both immature and mature cartilage (Fig. 1B). These results suggest that GPR68 was robustly expressed in both murine and human cartilage during course of chondrogenic differentiation in vivo and in vitro suggesting a potential role for GPR68 in articular cartilage. Owing to the robust and intensified expression of GPR68 among other members of proton sensing receptors in human and mice cartilage, we focus our study on GPR68 for in depth analysis of its functional role in cartilage during OA progression.

Figure 1 Expression profile of the proton sensing GPCRs genes.

Based on RNA-seq analysis of NCBI-GEO datasets-GSE114007 (human cartilage) and GSE110051 (mice cartilage) showing robust expression of GPR68. (A) Human cartilage derived from embryonic (5-week post conception), presumptive (17-week post conception), juvenile and adult stage; and (B) Immature and mature murine cartilage. The heatmap showed normalized Z-score value of read counts of genes. The raw dataset was normalized using Z-score along the columns and differential expressed proton sensing receptors with expression level greater than two-fold (FDR P value < 0.05) were shown as heatmap.

GPR68 expression is profoundly increased in human OA cartilage

To determine the role of GPR68 in OA pathogenesis, we assessed the expression level in human cartilage isolated from healthy donors and OA patients. OA cartilage showed macroscopic cartilage degeneration as compared to healthy cartilage and exhibited reduced proteoglycan content as analyzed by Safranin-O staining indicating the loss of cartilage ECM in OA patients (Fig. 2A). To investigate the comparative expression profile of GPR68, we analyzed RNA-seq data performed on performed on 38 individuals including 18 healthy and 20 OA human knee cartilage tissues (GSE114007) (Fisch et al., 2018). Our results as shown in Fig. 2B demonstrate that GPR68 is expressed in both healthy and OA cartilage and its mRNA levels are significantly increased in OA cartilage (N = 20) compared to healthy cartilage (N = 18) (# P < 0.01) (Fig. 2B). These results suggest that GPR68 might be playing a role OA pathogenesis.

Figure 2 GPR68 was robustly expressed in human OA cartilage.

(A) Saf-O-Fast green staining in healthy and OA cartilage indicating the severe loss of proteoglycans in OA; (B) Expression values of GRP68 (RKPM) in cartilage isolated from healthy and OA cartilage ***p ≤ 0.001, as compared to healthy cartilage based on RNA-seq analysis of datasets deposited in NCBI-GEO (GSE114007). The heatmap showed normalized Z-score value of read counts of genes. The raw dataset was normalized using Z-score along the columns.

GPR68 expression positively correlates with OA severity in human

To further confirm the role of GPR68 in OA pathogenesis, we analyzed its expression in human cartilage isolated from different stages of OA. High-grade OA cartilage (Mankin score ≥ 3;) showed severe cartilage degeneration as compared to low-grade OA cartilage (Mankin score ≤ 3) and exhibited loss of proteoglycan content as analyzed by Safranin-O and Fast green staining indicating the loss of cartilage ECM during OA progression (Fig. 3A). Our IHC and qPCR analyses showed that expression of GPR68 was significantly higher in cartilage with high-grade OA (N = 10) compared to the cartilage with low-grade OA (N = 10) (Fig. 3A, B; P < 0.01). The increased expression of the GPR68 positively correlated with the severity of OA further reinforcing our hypothesis that GPR68 plays a unique role in the pathogenesis of OA.

Figure 3 GPR68 expression correlates with OA severity.

(A) Saf-O/Fast green staining and IHC against GPR68 in low- and high-grade OA cartilage; (B) qPCR analysis for GRP68 expression in low- and high-grade OA cartilage *p ≤ 0.01, as compared to low grade OA cartilage.

Gpr68 expression is increased in murine cartilage of experimentally induced OA

To further confirm of role of Gpr68 in OA pathogenesis, we examined the expression level of GPR68 in murine cartilage of surgically induced OA. DMM surgery, which is transection of medial meniscotibial ligament, is a gold standard approach to model human post-traumatic OA (PTOA) (de Hooge et al., 2005; Ma et al., 2007) by inducing experimental OA in murine knee joint. This model reproduces most of the pathogenic events leading to PTOA following injuries that impact load distribution in the knee. Safranin-O/Fast Green staining of knee joints after 12-week post-DMM surgery demonstrated significant cartilage erosion and cartilage matrix depletion as shown by fibrillation of cartilage surface and reduced proteoglycan level in the medial femoral condyle and medial tibial plateau of DMM mice as compared to sham surgery (Fig. 4A). It has been used in dozens of previous studies including ours (Ansari et al., 2019), which aimed at determining the importance and roles of specific genes on OA initiation or progression. Moreover, IHC analysis using anti-GPR68 antibody in murine cartilage showed that protein levels of GPR68 were higher in DMM cartilage and meniscus as compared to sham cartilage (Fig. 4B). Interestingly, the meniscus cells in the histological section of DMM mice had stronger immunoreaction with anti-GPR68 antibody. However, as shown in Zoomed image (indicated by red arrows) increased immunoreactivity was observed in both cartilage and meniscus region of DMM mice as compared to sham control suggesting that GPR68 expression markedly increased during OA progression in mice. Further, negative antibody control which include the nonspecific rabbit IgG antibody used at same concentration did not show any immunoreactivity in the histological sections of murine joint (Fig. 4B). These results were further corroborated by transcriptome analysis of microarray dataset demonstrating increased Gpr68 expression in mice cartilage of DMM joints as compared to sham operated knee joints (GSE45793) (Bateman et al., 2013). (N = 4/group) (Fig. 4C; P < 0.01). Collectively, data support our hypothesis that Gpr68 play a significant role in cartilage erosion during OA progression in mice.

Figure 4 Gpr68 was robustly expressed in cartilage of experimental osteoarthritis in mice.

(A) Saf-O/Fast Green staining and (B) IHC analysis using anti-GPR68 and isotype control antibodies in the knee joints with DMM-induced OA as compared sham surgery; (C) Expression values of Gpr68 in cartilage derived from DMM and sham surgery mice based on Microarray analysis (GSE45793) (P < 0.01).

GPR68 DNA promoter shows hypomethylation in OA cartilage

DNA methylation plays a major regulatory role in gene transcription mainly via a negative regulation way (Ehrlich, 2019; Rauluseviciute, Drablos & Rye, 2020; Smith et al., 2020). The hypomethylated DNA facilitates the binding by transcription factors and promotes gene transcriptional activation (Ehrlich, 2019; Rauluseviciute, Drablos & Rye, 2020; Smith et al., 2020). To examine whether increased expression of GPR68 observed in OA cartilage was associated with decreased level of DNA methylation, we analyzed DNA methylome of cartilage in knee and hip OA. Genome-wide methylation was performed in chondrocyte DNA extracted from 12 high vs low grade OA cartilage using llumina Infinium Human Methylation450K BeadChip array (GSE63695) (Steinberg et al., 2017). We analyzed all 20 CpG sites for assessing the average DNA methylation levels of GPR68 in OA cartilage. Our analysis demonstrates the presence of total 5 CpG sites in the promoter (TSS1500 and TSS200) region of GPR68 (Fig. 5A). Interestingly, average promoter CpG methylation was significantly lower in high grade OA cartilage compared with the level in low grade OA cartilage (Fig. 5B, # P < 0.01). These results demonstrating hypomethylation of GPR68 promoter in severe OA cartilage are in concordance with increased GPR68 expression in high grade OA cartilage as shown in Fig. 3B. Taken together our data further highlights the role of GPR68 in OA progression.

Figure 5 GPR68 promoter was undermethylated in human OA cartilage.

(A) DNA-methylation profile of CpG sites in TSS1500, and TSS200 (promoter region) of GPR68 genes in healthy and OA cartilage; (B) Average promoter DNA methylation of GRP68 gene in healthy and OA cartilage based on Epigenomics data (Illumina 450K array) deposited in NCB-GEO (GSE63695) (P-value < 0.01).

GPR68 expression positively correlates with matrix degeneration in human and mice cartilage during OA progression

Since GPR68 expression robustly increased in high grade human OA cartilage and DMM operated knee joints in mice as shown in Figs. 3 and 4, we next determined whether GPR68 expression correlates with the degeneration of extracellular matrix. For this, we analyzed cartilage ECM content by Safranin-O staining which stain the sulfated GAG and proteoglycan levels of cartilage matrix in human cartilage explants and murine knee joint sections. Our analysis of human cartilage as shown in Fig. 3A revealed that as compared to low grade OA, high grade OA cartilage showed reduced proteoglycan content indicating the higher rate of matrix degeneration. Furthermore, IHC analysis of same cartilage explants showed that high grade OA cartilages exhibited increased immunoreactivity (Fig. 3A) demonstrating that during OA progression, human cartilage that displayed increased matrix degeneration showed positive correlation with protein level of GPR68. Similarly, during OA progression in mice, reduced Safranin-O staining indicating loss of proteoglycans in DMM operated mice knee joints at 12-week post-surgery also showed increased immunoreactivity with GPR68 antibody (Fig. 4B). These data further demonstrates that GPR68 level positively correlates with ECM degeneration in human and mice model indicating the plausible role of GPR68 in degeneration cartilage matrix during OA progression.

IL1β stimulation of primary human OA chondrocytes represents an in vitro model of OA

Enhanced production of matrix degrading proteases by activated chondrocytes is hallmark feature of OA and IL1β is key player of catabolism in OA joints which leads to progressive and irreversible cartilage degradation. To demonstrate IL1β as a key mediator of OA, we analyzed proteoglycan content as an indication of cartilage ECM degradation using Safranin-O and fast green staining of cartilage explant stimulated with or without IL1β. Our results demonstrate that IL1β stimulation of OA cartilage explants resulted in reduced Safranin-O staining, indicating loss of proteoglycans, in comparison to untreated explants demonstrating increased production of cartilage ECM degrading proteinases (Fig. 6A). To further demonstrate whether IL1β treatment induced matrix degrading protease gene expression in primary human OA chondrocytes, we prepared chondrocytes using full thickness cartilage using enzymatic digestion, cultured in monolayer, stimulated with IL1β (1 ng/ml) for 24 h, then assessed the expression profile of catabolic mediators mRNA levels using qPCR array. Our qPCR array data showed that IL1β treatment increased the mRNA expression of a plethora of catabolic genes, proinflammatory molecules, several inflammatory signaling genes (NF-κβ) and also reduced the expression of cartilage anabolic genes such COL2A1, ACAN etc., (Fig. 6B) in primary human OA chondrocytes. These data indicate that human chondrocyte cultures derived from pathologic cartilage samples produced matrix degrading protease repertoire mirroring that of OA joints, therefore monolayer culture of primary human OA chondrocytes stimulated with IL1β represent an ideal cellular model to study the pathogenesis of OA.

Figure 6 IL1β is a potent inducer of inflammation in monolayer culture of human OA chondrocytes.

(A) IL1β treatment induced significant loss of proteoglycans as demonstrated by Safranin O-Fast Green staining in human OA cartilage; (B) IL1β treatment induced potent inflammation in human OA chondrocytes as analyzed by increased mRNA levels of inflammatory and catabolic genes by qPCR array.

IL1β is a potent inducer of GPR68 expression in human OA cartilage and chondrocytes

Since IL1β is a key inducer of catabolic events in OA, we determined whether IL1β induced expression of GPR68 in primary chondrocytes. We prepared chondrocytes from knee cartilage of OA patients undergoing joint arthroplasty and monolayer cultures were stimulated with IL1β (1 ng/ml) for different time points. Our qPCR analysis showed that IL1β treatment induced the expression of GPR68 in a time dependent manner in human OA chondrocytes (Fig. 7A; P < 0.01). Consistent with the increased mRNA expression levels, immunostaining of histological sections of human cartilage revealed that the GPR68 protein level was also markedly high upon stimulation with IL1β (25 ng/ml, 72 h) (Fig. 7B). These results demonstrate that stimulation of both human chondrocytes and cartilage with IL1β, which is known to be associated with OA pathogenesis and cartilage matrix destruction, caused significant upregulation of the GPR68 mRNA and protein expression.

Figure 7 IL1β is potent inducer of GPR68 activation and Ogerin treatment repressed MMPs expression in human OA chondrocytes.

(A) IL1β (1 ng/ml) treatment significantly induced the expression of GPR68 in a time dependent manner (P < 0.01); (B) IHC against GPR68 in IL1β stimulated OA cartilage; (C) heatmap of mRNA expression of matrix degrading proteases MMP13, MMP3, MMP9, ADAMTS4, ADMATS5 genes in IL1β (1 ng/ml) treated OA chondrocytes; (D) MMP13 expression was positively correlated with the expression of GPR68 in IL1β treated OA chondrocytes; (E) Correlation analysis between expression of GPR68 and matrix degrading proteases in IL1β treated OA chondrocytes (P < 0.05).

IL1β induced GPR68 expression positively correlates with the expression of matrix degrading proteases in human OA chondrocytes

Our analysis demonstrates that IL1β is potent inducer of GPR68 expression in human OA chondrocytes and cartilage (Fig. 7A). We next determined whether IL1β induced expression of GPR68 correlates with the expression of matrix degrading proteases in human OA chondrocytes. We stimulated the monolayer culture of human OA chondrocytes with IL1β (1 ng/ml) for different time points and analyzed the gene expression of matrix metalloproteases such as MMPs and ADAMTSs. Our qPCR analysis revealed that as compared to untreated control, stimulation of OA chondrocytes with IL1β resulted in significantly high-level mRNA expression of several matrix degrading proteases such as MMP3, MMP9, MMP13, ADAMTS4 and ADAMT5 (Fig. 7C). These data indicate that IL1β is a potent inducer of GPR68 and proteases mRNA expression in human OA chondrocytes. Furthermore, OA chondrocytes that express high level of GPR68 under stimulation with IL1β showed positive correlation with expression level of MMP13 (Fig. 7D) and other matrix degrading proteases (Fig. 7E). Altogether, these data suggest that GPR68 may mediate catabolic and matrix degrading role of IL1β in human OA chondrocytes.

Ogerin mediated GPR68 activation repressed the expression of MMPs in human OA chondrocytes

Our correlation studies as shown in Figs. 7D and 7E demonstrate a clear correlation between mRNA level of GPR68 and MMPs in human chondrocytes. We next determined whether increased GPR68 activation modulate the expression of MMPs. To this end, we used small molecule activator of GPR68 in human chondrocytes and analyzed whether increased GPR68 had any effect on the expression of MMPs in presence of IL1β stimulation. Ogerin (OGR) is a synthetic GPR68 selective agonist, which activate GPR68 at physiological pH and enhanced the receptor activity (Huang et al., 2015). Human OA chondrocytes were treated with Ogerin (1–10 μM) for 2 h and then stimulated with IL1 β and mRNA expression of MMP3 and MMP13 was determined by SYBR Green gene expression assay. As reported previously (Khan et al., 2017a; Khan, Ansari & Haqqi, 2017), our data showed that treatment of OA chondrocytes with IL1β significantly increased the mRNAs expression of MMP3 and MMP13 as compared to untreated control OA chondrocytes (Figs. 7F and 7G). Interestingly, pretreatment of OA chondrocytes with Ogerin which in turn activates GPR68 significantly inhibited the mRNA expression of these matrix degrading proteases in a dose dependent manner (Figs. 7F and 7G). These results indicate that GPR68 activation by Ogerin in human chondrocytes exhibits anti-catabolic effects through repression of MMP3 and MMP13 mRNA expression under OA pathological conditions.

Discussion

GPR68 is a proton sensing orphan receptor which is activated upon binding to extracellular protons during acidosis or acidic stress. These pH sensors sense extracellular protons and stimulate different intercellular signaling pathways (Pereverzev et al., 2008; Tomura et al., 2008). Recent studies showed that GPR68 receptor is expressed in the intervertebral disc tissue (IVD) specifically in the endplate chondrocytes and involved in apoptosis induced by extracellular acid or IDD like acidic environment (Li et al., 2020; Yuan et al., 2014a). However, the role of GPR68 in articular cartilage is completely unknown. In this study, we observed GPR68 expression in musculoskeletal tissue in human and mice using various approaches including computational analysis including illuminating the druggable genome’ (IDG) and ‘PHAROS database’ (Hopkins & Groom, 2002; Nguyen et al., 2017; Rodgers et al., 2018), transcriptomics, epigenomics, quantitative PCR and immunohistochemical analysis in knee joint tissue, cartilage explants, and chondrocytes. Moreover, we found increased expression of GPR68 particularly in the damage and fibrillated areas of the human cartilage tissue from OA patients. Additionally, our results showed significant upregulation of Gpr68 expression, specifically in degenerated cartilage and inflamed meniscus of murine joint tissue during surgically induced OA model. Our data showed that GPR68 expression significantly increased in cartilage tissue isolated from OA patients with high grade disease severity showing a severe degeneration of extracellular cartilage matrix, when compared to cartilage from healthy volunteers or low-grade OA patients. Chondrocytes are the resident cell type present in cartilage. Our results showed that increased GPR68 expression in primary human chondrocytes stimulated with inflammatory cytokine IL1β which induced OA like pathogenic situation. Using several correlation analyses, we showed that GPR68 expression positively correlated with degree of OA severity, cartilage matrix degeneration and expression of MMP13 and other matrix degrading proteases in human chondrocytes. Consequently, we hypothesize that GPR68 activity leads to an increased catabolic gene expression which triggers the activation of signaling cascades involved in further ECM degeneration and degradation of cartilage tissue leading to OA progression. Using selective activation of GPR68 by the treatment with small molecule agonist, the Ogerin, our results further showed that increase GPR68 activity inhibits the expression of matrix degrading proteases under OA like inflammatory conditions in human chondrocytes in vitro.

Osteoarthritis is the most common form of arthritis among older adults. OA is a degenerative joint disorder characterized by the breakdown of cartilage, leading to joint pain, stiffness, and loss of function. It is also one of the most common causes of physical disability among adults (Goldring & Berenbaum, 2004; Goldring & Marcu, 2009; Goldring et al., 2008). Inflammation and matrix degeneration are considered as major mediators of OA. Degeneration of extracellular matrix triggers the activation of signaling pathways leading to inflammation and apoptosis of chondrocytes during OA (Ferguson et al., 2018; Khan et al., 2017b). Several recent studies showed that inflammatory cytokine IL1β induced inflammation which activates the production of matrix degrading proteases leading to degeneration of ECM during OA progression. Recent studies have shown the role of GPR68 in the regulation of inflammatory responses and activation of GPR68 had been shown to increase intestinal inflammation, fibrosis, and colitis in experimental model (de Valliere et al., 2021; de Vallière et al., 2015; de Vallière et al., 2022). We hypothesize that GPR68 might be playing a role in inflammation and subsequent activation of matrix degeneration during OA progression. A recent study using end plate chondrocytes which are primarily avascular tissue like articular chondrocytes demonstrated the role of GPR68 in acid induced apoptosis, indicating the role of GPR68 in pathogenesis of IDD (Yuan et al., 2014a). We employed in vitro model of cartilage degeneration using human OA cartilage explants and in vivo model of surgically induced OA in mice to demonstrate the role of GPR68 in the degradation of extracellular matrix in OA pathogenic condition. Our results elucidated the positive correlation of GPR68 expression with loss of proteoglycan content in human OA cartilage and surgically induced OA in mice indicating the plausible role of GPR68 in OA pathogenesis.

During OA progression, cartilage tissue is damaged due to mechanical injury, trauma, aging, and inflammation. Degeneration of ECM is hallmark of OA pathogenesis (Khan et al., 2017a; Khan, Ahmad & Haqqi, 2018; Khan, Ansari & Haqqi, 2017; Khan et al., 2017b; Khan et al., 2017c; Khan et al., 2023b). Extracellular matrix degradation in OA primarily occurred by activation of matrix degrading proteases such as aggrecanases and metalloproteinases (MMPs) (Goldring, 1999; Goldring & Berenbaum, 2004; Goldring & Marcu, 2009). MMP13, MMP3, MMP9, ADAMTS4 and ADAMTS5 are produced by chondrocytes in response to IL1β that contribute to OA pathogenesis. Therefore, we investigated the effect of GPR68 expression on the IL1β induced expression of these proteases in human OA chondrocytes. Our results showed that chondrocytes upon stimulation with IL1β showed a robust increase in the expression of GPR68 mRNA and induced GPR68 expression demonstrated the repression of mRNA expression of several matrix metalloproteinases such as MMP13, and MMP3. Together, these results demonstrate a new role for a proton sensing orphan gene in the regulation of MMPs expression in chondrocytes during disease progression.

Activation of inflammatory signaling has been shown to induce matrix degeneration in several degenerative disorders including OA (Goldring, 1999; Goldring & Berenbaum, 2004; Goldring & Marcu, 2009; Loeser, Collins & Diekman, 2016). Several in vitro studies including ours showed that IL1β stimulation of articular chondrocytes leads to expression, and secretion of active MMP13 which cleaves the major components of ECM including type two collagen and aggrecan (Goldring, 1999; Goldring & Berenbaum, 2004; Goldring & Marcu, 2009; Khan et al., 2017b; Khan, Ahmad & Haqqi, 2018; Loeser, Collins & Diekman, 2016). Our results presented in this study showed that IL1β stimulation of OA chondrocytes induced the expression of GPR68 and several proteases including MMP13 and MMP3 in human OA chondrocytes. To corroborate our hypothesis that GPR68 plays a role in matrix degeneration and activation of GPR68 mediates the catabolic effect of IL1β, we performed gain-of function studies in chondrocytes by using a selective positive allosteric modulator of GPR68. Recently, small molecule Ogerin was identified as a positive allosteric modulator (PAM) of GPR68, which activates GPR68 at physiological pH and enhanced proton receptor activity by induction of Gαs signaling pathway (Huang et al., 2015). Using our established IL1β induced in vitro model of OA, our results demonstrate the protective role of Ogerin and showed that GPR68 activation exhibited anti-catabolic effect by repressing the expression of matrix degrading proteases including MMP3 and MMP13. Human chondrocytes used in our study were derived from OA cartilage where significant matrix degeneration has already occurred, the ability of Ogerin to repress matrix degrading protease gene expression has significant implications to halt or slow down the cartilage erosion and thus halt the disease progression.

Although data present here provide a clear role of GPR68 in the regulation of matrix degradation during OA progression. However, an in-depth mechanistic analysis of using genetic manipulation of GPR68 using in vitro and in vivo GPR68-/-mice are needed to determine the specific role of GPR68 in the transcription regulation of matrix degrading proteases genes during OA progression. We envision a multitude of follow-up studies using overexpression and knockout of GPR68 mice model aimed at thoroughly examining the role of GPR68 during OA progression in aging and post traumatic surgery model which will define the precise role of GPR68 in controlling the cartilage degeneration. Our future studies on the expression of GPR68 during aging and post-traumatic OA are likely to shed new light on the role of GPR68 in regulation of catabolic gene expression and pathogenesis of OA.

Although this study provides first demonstration of role of proton sensing GPR68 in OA, there are some limitations of this study. Our studies are mainly correlative in nature and focusses on human chondrocytes in vitro, it is important to understand the phenotypes of in vivo GPR68 activation or GPR68 loss of function to accurately interpret its role in matrix degeneration. Additionally, our in vitro culture condition used physiological pH, owing to fact that GPR68 activity is higher under acidic condition, further studies are needed to determine the effect of pH on the activation of GPR68 in chondrocytes and its role in the regulation of ECM degeneration in cartilage explants. Future work investigating the effect of Ogerin in joint instability model of rat and mice in vivo will further expand our understanding of GPR68 in pathological states which will open a new avenue for potential of Ogerin as therapeutics in OA.

Conclusion

In conclusion, the present study is the first to demonstrate that a proton sensing orphan gene GPR68 is expressed in human and mice cartilage. Our results demonstrated that GPR68 expression profoundly increased in cartilage and meniscus during surgically induced OA in mice. GPR68 expression in human and mice cartilage correlates with matrix degeneration and severity of OA progression. Additionally, stimulation of chondrocytes with inflammation results in induced expression of GPR68 and increased GPR68 activation further repressed the expression of matrix degrading proteases including MMP13 and MMP3 in OA chondrocytes indicating the plausible role of GPR68 in regulation of metalloproteinase gene expression. These results identify GPR68 as a possible therapeutic target in the regulation of matrix degradation during OA.

Supplemental Information

Supplemental Information 1 Raw data related to Figures 2–7.

Click here for additional data file.

Supplemental Information 2 ARRIVE 2.0 Checklist.

Click here for additional data file.

Additional Information and Declarations

Competing Interests

Author Contributions

Ethics

Data Availability

The authors declare that they have no competing interests.

Nazir M Khan conceived and designed the experiments, performed the experiments, analyzed the data, prepared figures and/or tables, authored or reviewed drafts of the article, and approved the final draft.

Martha E Diaz-Hernandez performed the experiments, analyzed the data, prepared figures and/or tables, authored or reviewed drafts of the article, and approved the final draft.

William N Martin performed the experiments, analyzed the data, authored or reviewed drafts of the article, and approved the final draft.

Bhakti Patel performed the experiments, prepared figures and/or tables, and approved the final draft.

Samir Chihab performed the experiments, analyzed the data, authored or reviewed drafts of the article, and approved the final draft.

Hicham Drissi conceived and designed the experiments, authored or reviewed drafts of the article, and approved the final draft.

The following information was supplied relating to ethical approvals (i.e., approving body and any reference numbers):

The Emory IRB approved the study (IRB ID STUDY00003990).

The following information was supplied regarding data availability:

The raw data are available in the Supplemental File.

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
