# Peer review of "pH-sensing G protein-coupled orphan receptor GPR68 is expressed in human cartilage and correlates with degradation of extracellular matrix during OA progression"

_PeerJ, doi:10.7717/peerj.16553_

## Round 0.1 · original submission · Major Revisions

There are some major concerns raised by reviewers. Mechanistic analyses need to be incorporated into the paper.

(1) Please ask Hicham to supply their email in their PeerJ profile here: <https://peerj.com/settings/details/#basic-info>;
(2) Before you complete the revisions, please also update Hicham’s email here <https://peerj.com/manuscripts/85341/authors/>;
(3) Lastly, please explain in your response letter why mohd.n.khan@emory.edu was used for this author.

Reviewer 1 ·

Basic reporting

no comment

Experimental design

no comment

Validity of the findings

no comment

Additional comments

This manuscript by Nazir Khan et al. investigates the expression of GPR68 in OA. However, in my opinion, data presented in this study does not fully support the conclusions drawn.
1. Discrepancy in sample size between author analyzed RNA-seq data and GSE114007 data including 18 healthy and 20 OA human knee cartilage tissues (Fisch et al.,2108). Whereas author analyses were performed with 18 samples (n=8 healthy and n=10 OA). This makes the analyzed data unconvincing or biased. What is the justification for this sampling strategy?
2. The specificity of GPR68 antibody was not verified in the immunohistochemical analysis of GPR68 in human and mice cartilage. There was also lacking quantification of IHC experiment, and GPR68 expression appeared to be no different between the sham and DMM groups in Figure 3A.
3. DMM surgery can cause cartilage destruction, however, the author results showed no cartilage destruction as assessed by Safranin-O staining 12 weeks post DMM surgery, and no OA score was performed, so it is difficult to judge the success of the surgery and make results are unconvincing.
4. Authors have omitted crucial details (e.g. qPCR primer sequence) from materials and methods, which make it difficult to assess the validity of the results or repeat the experiments by independent researchers.
5. Introduction needs more reasonable. The expression and function of GPR68, G2A, and GPR65 in cartilage during OA are unknown, why only GPR68 was studied.
6. Poor picture quality and lack of scale bar.
7. The results description is not precise, leading to misunderstanding among readers. e.g. line 312 N=18 is 8 healthy cartilage and 10 OA cartilage, instead of each group containing 18 samples. The same problem for Line 323.
8. By analyzing RNA-seq dataset, authors confirmed that GPR68 is highly expressed in different stages of chondrogenic differentiation. Notably,GPR68 was also highly expressed in embryo and immature cartilage, which contradicts GPR68 is highly expressed in OA.

·

Basic reporting

The paper titled "pH-sensing G protein-coupled orphan receptor GPR68 is expressed in human cartilage and correlates with degradation of extracellular matrix during OA progression" investigates the expression and function of the pH-sensing G protein-coupled receptor GPR68 in human cartilage during the progression of osteoarthritis (OA). The authors aim to determine whether GPR68 is involved in the regulation of catabolic gene expression using an in vitro model of OA catabolism. This is a potentially exciting insight into osteoarthritis pathology but would benefit from a little more clarity in experimentation.

The article is written in clear English and claims are very very specific. The references are sufficient and the background is clear. All together this paper is well put together.

Experimental design

1) The study ends with a mechanistic interrogation of GPR68 acting downstream of IL1B, however, it not very rigorous given the correlative nature of the study. Performing the same experiments in the setting of GPR68 knockdown or chemical modulation (Ogerin/Isoxazole as agonists or Ogermorphin (Bioarxiv) as an antagonist).

Validity of the findings

1) Figure 1 is unclear, as to what the heat map is showing please provide information on normalization.
2) Figure 3 would benefit from qpcr of normal bone as a comparison if possible. This is particularly important as it is unclear if the data from RNAseq Figure 2b is driven by the extreme cases of OA. In other words, this reviewer in not convinced that GPR68 is increased in low grade OA vs Control.
3) The IHC for GPR68 in Figure 4a is not convincingly increased in OA model compared to sham. However, the data from microarray suggests otherwise. Confirmation of the data seen in microarray with QPCR would be nice, quanification of GPR68 staining would also be nice.

Reviewer 3 ·

Basic reporting

The authors provide a compelling report between the association of GPR68 expression and ECM degradation in cartilage.
I will keep my comments quite brief as I find only one major concern.
the manuscript is clear and well written. It is overall relatively descriptive, describing an association between GPR68 and cartilage degradation. Although they eloquently define that GPR68 expression is induced by IL1-B, the mechanisms beyond that are incompletely explored. They reference the use of GPR68 KO mice in the discussion. A requirement of their use for this manuscript is likely beyond reason for this paper. However, the use of GPR68 knockdown and overexpression in chondrocytes should be performed to more clearly define a direct link between GPR68 and cartilage destruction. siRNA and plasmids are commercially available so it is not clear why the authors did not explore these experiments.

Experimental design

strong with the exception of the lack of genetic manipulation of GPR68

Validity of the findings

valid

Additional comments

should genetic manipulation experiments confirm the strong data linking GPR68 to cartilage destruction, I would recommend acceptance of the article

Reviewer 4 ·

Basic reporting

Thanks for giving me the opportunity to review the manuscript, focusing on the role of the pH-sensing G protein-coupled receptor (GPCR), GPR68, in osteoarthritis (OA) progression and cartilage degradation.
I have given some feedback that can help in the improvement of the manuscript:

1: There are several instances where the language can be improved for better readability and comprehension and a thorough proofreading should be done to correct any grammatical errors and spelling mistakes. For example: When using acronyms or abbreviations, they should be defined correctly (matrix degrading proteases (MMP13) or ADAMTSs) and maintain consistency in terminology and style throughout the text (check GPR68 and Gpr68) (check IL-1β and IL1β).
2: Introduction part needs major revision in grammar and structure. The details of results should not be described in introduction and only the hypothesis of study should be mentioned.
3: The study's methodology is commendable; however, the discussion section requires significant restructuring to avoid repetition and improve its coherence. It should follow a structured format as follows:
Introduction of Findings (clearly presenting the key findings of the study).
Comparison with Similar Studies (compare the study's results with relevant research that has reported similar findings and provide mechanistic explanations to justify the similarities).
Addressing Inconsistent Results (discuss any discrepancies or inconsistencies between the study's findings and those of other studies and offer possible reasons for these variations).
Strengths of the Study (highlight the strengths of your current study).
Limitations (discuss the limitations of the study, such as potential biases, small sample size or experimental constraints).

Experimental design

No comments.

Validity of the findings

No comments.

---

## Round 0.2 · accepted · Accept

The authors have addressed the reviewers' concerns.

·

Basic reporting

The text is clearly written and revisions have clarified uncertainties.

Experimental design

No comment.

Validity of the findings

Findings are novel and well supported.

Reviewer 3 ·

Basic reporting

Clear

Experimental design

The Research question is well-defined, relevant & meaningful.
Experiments are well-designed

Validity of the findings

findings are valid

Additional comments

none